# Spherical Aberration and Accommodative Insufficiency: Is There a Link?

Jessica Gomes, Kishor Sapkota and Sandra Franco *

Centre of Physics, School of Sciences, Campus de Gualtar, University of Minho, 4710-057 Braga, Portugal
* Correspondence: sfranco@fisica.uminho.pt

**Abstract:** Given the relationship between spherical aberrations and accommodation, the study of these aberrations can be helpful to understand accommodative response in subjects with accommodative dysfunctions. The purpose of this study was to evaluate on-axis and off-axis changes of primary and secondary spherical aberrations, $Z(4,0)$ and $Z(6,0)$, with accommodation in subjects with accommodative insufficiency (AI). Ten subjects with AI and eleven without any accommodative dysfunction (control) participated in this study. On-axis defocus $Z(2,0)$, $Z(4,0)$, and $Z(6,0)$ were obtained in both groups with a Hartmann–Shack aberrometer for the unaccommodated state and with 1.00 D, 2.44 D, 3.83 D, and 4.73 D of accommodative stimuli. $Z(4,0)$ and $Z(6,0)$ were also measured on 11.5° and 23° temporal, nasal, superior, and inferior retinal areas for unaccommodated state and for 2.44 D of accommodative stimulus. In the control group, $Z(4,0)$ became more negative with accommodation and $Z(6,0)$ became more positive, as was expected according to previous studies. This tendency was not observed in the group of subjects with AI group for $Z(4,0)$ or for $Z(6,0)$. No differences on off-axis $Z(4,0)$ and $Z(6,0)$ were observed between the groups. The changes of spherical aberrations with accommodation seem different in subjects with AI compared to those without any accommodative dysfunction. Those with AI do not present a decrease in $Z(4,0)$ and an increase in $Z(6,0)$ with accommodation as occurs in eyes without this type of dysfunction. Understanding how the optics of the eye changes with accommodation can be helpful to understand the origin of accommodative dysfunctions.

**Keywords:** spherical aberration; accommodation; accommodative insufficiency





## 1. Introduction

Accommodation is the ability of the crystalline lens to change its shape and power to focus on the retina objects at different distances [1].

Ocular accommodation is related to several factors, and one of them is the optical quality of the eye. The changes in the crystalline lens during accommodation cause changes in the optical quality of the eye and can potentially contribute to the accuracy of the accommodative response (AR) [2,3]. According to the literature, primary ($Z(4,0)$) and secondary ($Z(6,0)$) spherical aberrations are closely related to the accommodation process [4–6]. Several studies showed that, in general, $Z(4,0)$ is positive in the relaxed state and becomes less positive or more negative when accommodation increases, reaching zero between 2.00 D and 3.00 D of accommodation [4–12]. Moreover, when positive $Z(4,0)$ is added, the accommodative lag increases, whereas when negative $Z(4,0)$ is induced, accommodative lag decreases [13].

On the other hand, $Z(6,0)$ tends to be more positive as the AR increases [9–11] and, at the same time, the effect of $Z(6,0)$ tends to increase AR [9,10,12].

Beyond central aberrations, peripheral optics have aroused interest over the last years [14–16]. However, the effect of accommodation on off-axis high order aberrations (HOA) has not been studied in depth. A previous study revealed that off-axis $Z(4,0)$ becomes more negative with accommodation in myopic eyes but not in emmetropes [16].

Lundstrom et al. [17] found a difference in the effect of accommodation on peripheral aberrations in myopic and emmetropic subjects. Another recent study revealed significant differences on peripheral HOA between unaccommodated state and with 2.50 D of accommodation stimulation [18].

When a non-presbyopic subject is not able to sustain focus for near vision, it is called accommodative insufficiency (AI) [19]. In this kind of dysfunction, for reasons still unknown, the eye presents amplitude of accommodation (Ac) below the lower limit of the expected for the patient's age, and generally also has low accommodative facility and an increased accommodative lag [20–23]. Individuals with AI report several symptoms, such as blurred near vision, headache, and eyestrain [21,22,24,25]. Furthermore, AI decreases academic performance, as these individuals avoid reading and present fatigue and difficulty with continuous reading [22].

Although the wide range of the prevalence of AI reported in the literature, studies suggested that it is commonly found in the population [20,26]. Children are the subjects more affected by this disorder, reaching a prevalence of 61.6% between 6 and 16 years old [25]. High school students and university students are also affected by AI, with prevalence between 4.1% and 6.2% [27–29].

Given the close relationship between primary and secondary spherical aberrations and ocular accommodation, it is important to understand whether this behaviour also occurs in subjects with AI and what their role is in the control of AR and in the development of these anomalies.

The hypothesis of this preliminary study is that the changes of spherical aberrations during accommodation might be different in subjects with accommodative insufficiency. Therefore, its purpose was to analyse central and peripheral spherical aberrations and how they change with accommodation in subjects with accommodative insufficiency compared to those without this dysfunction. The aim was to understand if these ocular aberrations can affect or be affected by accommodation in subjects with AI.

## 2. Materials and Methods

### 2.1. Subjects

The selection of subjects to participate in this study was based on the results of an optometric visual examination that included the measurement of visual acuity (VA) at far and near vision, static retinoscopy, subjective refraction, and accommodative and binocular vision exams. Accommodative exams included measurement of Ac by the Sheard method, monocular facility of accommodation (MFA) for near vision with +2.00/−2.00 D lenses, and accommodative lag by monocular-estimated method (M.E.M.) retinoscopy.

Individuals with Ac 2.00 D below minimum age-based norms as defined by Hofstetter's formula (15–0.25 × age (years)) and accommodative lag above +0.75 D were considered with AI [19]; the control group presented Ac, accommodative facility, and lag within the expected range.

The enrolled participants were then grouped into two different groups: 10 subjects with AI and a control group with 11 subjects without any accommodative dysfunction (control).

The control group had a mean (±standard deviation) age of 24.18 ± 3.16 years old, mean spherical equivalent +0.13 ± 0.37 D, and astigmatism less than 1.75 D. The AI group had a mean age of 21.20 ± 2.25 years old, mean spherical equivalent +0.03 ± 1.07 D, and astigmatism less than 1.75 D. In both groups, the best-corrected visual acuity was greater or equal to 20/20. None of the subjects presented a history of ocular pathology or pathologies that could affect vision, ocular surgery, or orthokeratology, and they have not received any treatment for the AI.

The study adhered to the tenets of the Declaration of Helsinki and was approved by the Ethical Sub commission of Life and Health Science of University of Minho. All subjects signed an informed consent with the explanation of the procedures.

### 2.2. Experimental Procedure

An adaptive optical system with an in-house Hartmann–Shack aberrometer (Thorlabs WF150-7AR) (Figure 1) was used to measure ocular aberrations on the subject's right eye, while accommodation was stimulated with negative lenses in the same eye using a motorized system (MS). The aberrometer had a resolution of 1280 × 1024 and 39 × 31, with lenses working at a frequency of 15 Hz. The power of the super luminescent diode (SLD), used to generate the optical beam, was 10 µW (L8414-04, Hamamatsu, Shizuoka, Japan) at the eye and had a spectral maximum at 830 nm. The beam diameter in the wavefront sensor was around 4 mm and the effective diameter used for measuring the aberrations was 2 mm.

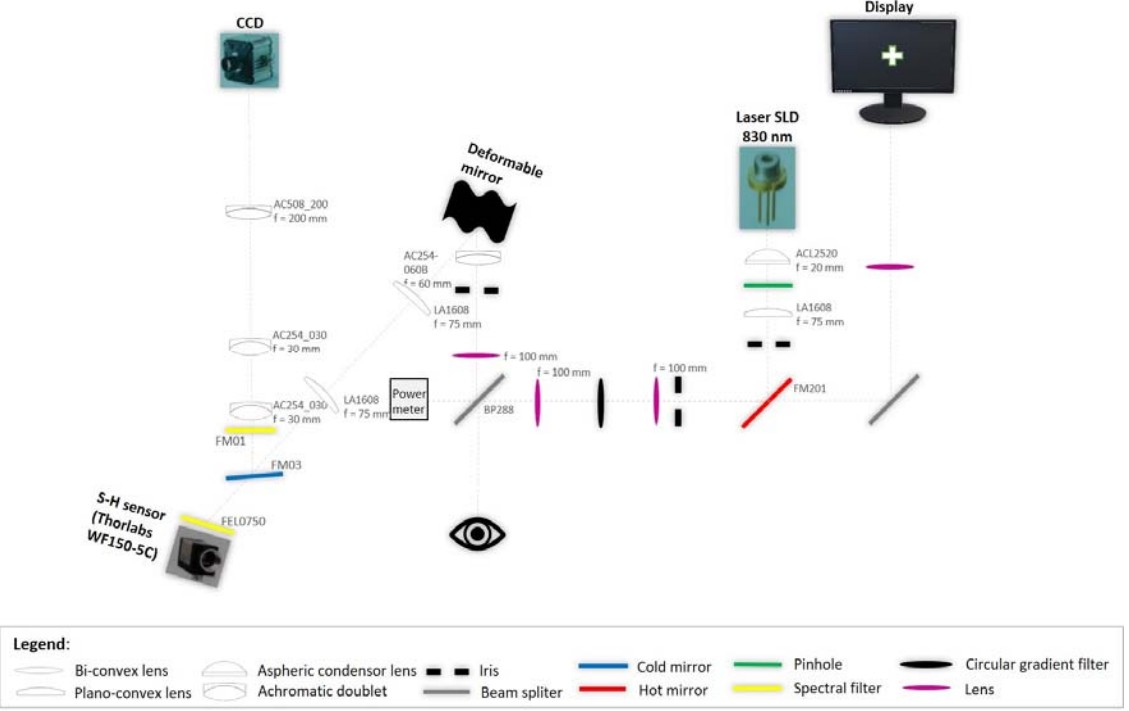

**Figure 1.** Schematic representation of the set-up.

The subject's eye was aligned with the system and stabilized with the help of a chin and forehead support unit. The target was a white cross over a black background at a simulated distance of 6 m and the subjects were asked to maintain it as clearly as possible. The subject's refractive error has been fully corrected (sphere and astigmatism) with lenses during measurements and all procedures were performed by an experienced specialist.

### 2.3. On-Axis Aberrometry

A motorized system (MS), controlled by a software and synchronized with the ocular wavefront aberration measurement, had been previously developed in the Centre of Physics of University of Minho [30], and where different negative lenses could be placed in front of the subjects' eye to stimulate the accommodation and simultaneously measure ocular aberrations. The acquisition of the ocular aberrations was synchronized with the changes in the accommodative stimulus.

The subject was asked to fix the target while accommodation was stimulated and measured central (on-axis) ocular aberrations in real-time. The measurements were taken every 100 milliseconds in mesopic conditions and without the use of mydriatic.

Every subject was submitted to different cycles of accommodation and disaccommodation using a negative lens in the following order: 0 D → −1.00 D → −2.50 D → −4.00 D → −5.00 D. Each lens was placed in front of the subject's eye for about 5 s, obtaining approximated 50 measures for each accommodative stimulus (AS), and the average was

calculated. Due to the distance from the lenses to the eye (20 mm) and considering the distance to the target, the effective AS for each lens is presented in Table 1.

**Table 1.** Lenses used to stimulate accommodation and its effective AS.

| Lens (D) | Effective AS (D) |
|---|---|
| −1.00 | 1.00 |
| −2.50 | 2.44 |
| −4.00 | 3.83 |
| −5.00 | 4.73 |

With the lenses −4.00 D and −5.00 D it was only possible to measure in 7 and 10 subjects, respectively, because of the small pupil size due to its constriction during the accommodative process [31] in some subjects, which did not allow for the system to accurately measure wavefront aberrations.

AR was obtained for 2.25 mm pupil radius from defocus aberration Z(2,0) by Equation (1), considering the lens in front of the subjects' eye [32,33].

$$AR = \frac{-4\sqrt{3}Z_0^2}{r^2},\qquad(1)$$

where $r$ is the pupil radius and $Z_0^2$ is the defocus aberration of the subject.

*2.4. Off-Axis Aberrometry*

To measure off-axis ocular aberration, the cross was moved 11.5 and 23 degrees off-axis in the temporal, nasal, superior, and inferior visual fields. The amount of eccentricity was limited within 23 degrees on each direction because it was not possible to obtain data with further eccentricity on some subjects with the setup. The process was then repeated with the minus lens −2.50 D placed in front of the eye (demand needed for daily near vision tasks, which is usually at 40–50 cm) to obtain data on peripheral retinal areas for an accommodation stimulus of 2.44 D.

*2.5. Statistical Analysis*

Statistical analysis was performed in version 4.1.3 of the software R.

The normality of the data was tested using the Shapiro–Wilk test and the results were compared between the control and AI groups using the t-test and the Mann–Whitney test for parametric and non-parametric data, respectively. It was considered statistically significant when $p < 0.05$.

**3. Results**

The results of accommodative exams performed before aberrometry are shown in the Table 2. Ac, M.E.M., and MFA were significantly lower in the AI group than in the control group.

**Table 2.** Mean and standard deviation of Ac, M.E.M., and MAF of the control and AI groups.

| | Ac (D) | M.E.M. (D) | MFA (cpm †) |
|---|---|---|---|
| Control | 9.43 ± 1.15 | 0.64 ± 0.17 | 14.64 ± 4.15 |
| AI | 5.7 ± 1.14 | 0.88 ± 0.24 | 7.3 ± 4.35 |
| *p*-value | <0.01 * | <0.01 * | <0.01 * |

* Statistically significant; † cpm = cycles per minute.

Mean values of AR for each AS are represented in Table 3. As expected, ARs were lower for the AI group compared to the control group, with differences being statistically significant for the stimuli 1.00 D ($p < 0.001$) and 2.44 D ($p < 0.001$).

**Table 3.** Mean AR and standard deviation of the control and AI groups for the different AS.

| | Mean AR (D) | | | |
|---|---|---|---|---|
| AS (D) | 1.00 | 2.44 | 3.83 | 4.73 |
| Control | +1.02 ± 0.36 | +2.40 ± 0.84 | +3.90 ± 0.89 | +5.10 ± 0.44 |
| AI | +0.25 ± 0.16 | +1.20 ± 0.80 | +2.95 ± 0.21 | +3.99 ± 1.67 |
| *p*-value | 0.016 * | 0.014 * | 0.115 | 0.368 |
| N | 21 (11/10) | 21 (11/10) | 17 (10/7) | 10 (4/6) |

* Statistically significant; N: number of subjects (Control/AI).

### 3.1. On-Axis Aberrometry

Figure 2 shows the mean values of Z(4,0) and Z(6,0) as a function of AR in the control group. Z(4,0) tends to become more negative (or less positive) and Z(6,0) tends to be more positive as the AR increases. On the other hand, in the AI group, this tendency was not observed for either of the two spherical aberrations (Figure 2). Z(4,0) showed even more positive values in accommodated states than in an unaccommodated state, and Z(6,0) remained constant.

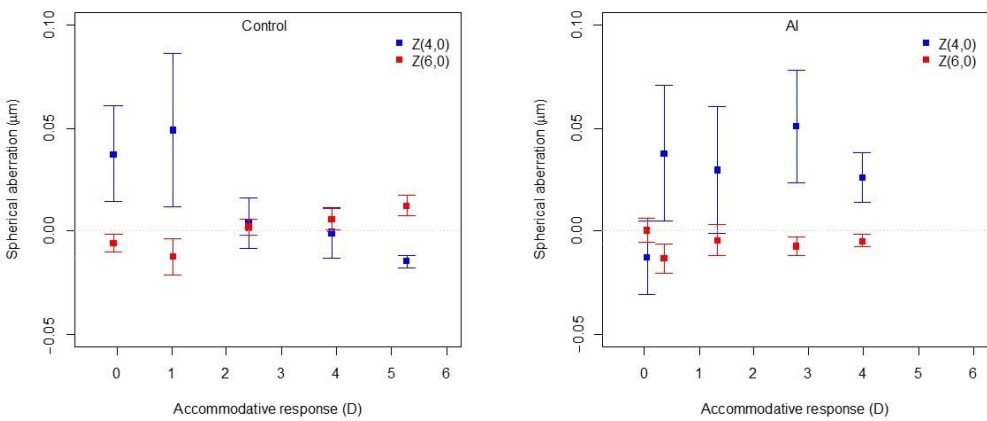

**Figure 2.** The mean of spherical aberrations Z(4,0) and Z(6,0) as a function of the accommodative response in the control and AI groups, with standard error bars.

### 3.2. Off-Axis Aberrometry

Mean values of off-axis Z(4,0) and Z(6,0) in both groups for unaccommodated and accommodated states are shown in the Table 4. With the AS of 2.44 D, temporal Z(6,0) at 11.50° was significantly more negative in the subjects with AI than in the control group ($p < 0.05$). No more significant differences were found in the peripheral retina between the control and AI groups. The changes of Z(4,0) and Z(6,0) caused by accommodation, i.e., the differences between accommodated and unaccommodated states in the different axis were also compared between the groups, but no statistically significant differences were found.

**Table 4.** Mean Z(4,0) and Z(6,0) and standard deviation in the unaccommodated and accommodated states for both groups in the 11.5° and 23° temporal (T), nasal (N), superior (S), and inferior (I) retina. *p*-value corresponds to the difference between groups.

| | Z(4,0) | | | Z(6,0) | | |
|---|---|---|---|---|---|---|
| | Unaccommodated | | | | | |
| | Control (µm) | AI (µm) | *p*-Value | Control (µm) | AI (µm) | *p*-Value |
| T 11.5° | −0.007 ± 0.124 | 0.005 ± 0.076 | 0.941 | 0.009 ± 0.168 | 0.081 ± 0.175 | 0.882 |
| T 23° | 0.114 ± 0.181 | 0.071 ± 0.165 | 0.400 | 0.026 ± 0.129 | 0.041 ± 0.171 | 0.604 |
| N 11.5° | 0.071 ± 0.135 | 0.065 ± 0.125 | 0.900 | 0.004 ± 0.099 | −0.047 ± 0.064 | 0.370 |
| N 23° | 0.073 ± 0.145 | 0.004 ± 0.099 | 0.249 | 0.016 ± 0.151 | 0.026 ± 0.095 | 0.604 |
| S 11.5° | <0.001 ± 0.164 | 0.063 ± 0.107 | 0.340 | −0.012 ± 0.072 | −0.040 ± 0.089 | 0.455 |
| S 23° | −0.022 ± 0.165 | 0.070 ± 0.138 | 0.207 | −0.091 ± 0.199 | −0.031 ± 0.205 | 0.400 |
| I 11.5° | 0.036 ± 0.146 | 0.042 ± 0.088 | 0.710 | 0.077 ± 0.225 | 0.014 ± 0.252 | 0.563 |
| I 23° | 0.045 ± 0.102 | 0.013 ± 0.042 | 0.375 | <0.001 ± 0.107 | 0.024 ± 0.180 | 0.549 |

**Table 4.** *Cont.*

| | Z(4,0) | | | Z(6,0) | | |
| --- | --- | --- | --- | --- | --- | --- |
| | | | **Accommodated** | | | |
| | **Control (μm)** | **AI (μm)** | ***p*-Value** | **Control (μm)** | **AI (μm)** | ***p*-Value** |
| T 11.5° | $0.024 \pm 0.054$ | $0.004 \pm 0.012$ | 0.941 | $0.001 \pm 0.042$ | $-0.025 \pm 0.043$ | 0.031 * |
| T 23° | $-0.034 \pm 0.034$ | $-0.018 \pm 0.033$ | 0.309 | $-0.004 \pm 0.042$ | $-0.015 \pm 0.055$ | 0.638 |
| N 11.5° | $-0.019 \pm 0.040$ | $-0.009 \pm 0.039$ | 0.603 | $-0.004 \pm 0.026$ | $-0.007 \pm 0.028$ | 0.783 |
| N 23° | $-0.024 \pm 0.062$ | $-0.009 \pm 0.036$ | 0.900 | $-0.015 \pm 0.038$ | $0.017 \pm 0.039$ | 0.092 |
| S 11.5° | $-0.012 \pm 0.057$ | $-0.008 \pm 0.027$ | 0.503 | $-0.007 \pm 0.016$ | $0.016 \pm 0.023$ | 0.400 |
| S 23° | $-0.006 \pm 0.046$ | $-0.010 \pm 0.041$ | 0.838 | $0.025 \pm 0.067$ | $0.017 \pm 0.061$ | 0.779 |
| I 11.5° | $-0.004 \pm 0.027$ | $-0.010 \pm 0.036$ | 0.624 | $-0.022 \pm 0.063$ | $0.014 \pm 0.082$ | 0.286 |
| I 23° | $-0.008 \pm 0.034$ | $0.008 \pm 0.028$ | 0.370 | $0.012 \pm 0.030$ | $-0.012 \pm 0.051$ | 0.285 |

T = Temporal; N = Nasal; S = Superior; I = Inferior. * Statistically significant between groups.

## 4. Discussion

Several authors defined AI as an inability to stimulate accommodation, presenting lower values of Ac than expected for age, among other findings, such as low facility of accommodation and high accommodative lag [19–21].

Accommodative response was obtained by defocus for different AS in both groups. The ARs were lower for all stimulus in the AI group than in the control group, although the statistically significant differences were only observed for the stimulus of 1.00 D and 2.44 D. The greater dispersion found for the highest accommodative demand may explain why statistically significant differences were found only for the two lowest stimuli. Moreover, the smaller sample size for the two highest accommodative stimuli can justify the lack of statistically significant differences. However, even without statistically significant differences found for the two highest stimuli, the mean accommodative lag was about 1.00 D lower than in the control group.

The results of this study showed that there are differences in Z(4,0) and Z(6,0) with ocular accommodation between the groups.

Several previous studies reported that Z(4,0) become more negative as AR increases, and in general it is positive for the unaccommodated state and reaches zero between 2.00 D and 3.00 D of accommodation [4–11]. In this study, the same results were found in the control group, with a negative tendency between Z(4,0) and accommodation. In contrast, this tendency was not observed in the subjects with AI, where Z(4,0) average was negative in the unaccommodated state and tends to keep the same value for the different AR or became even more positive.

As spherical aberration plays an important role in the control of accommodation, the sign and amount of this aberration can influence the accuracy of AR [8,12,13,34]. Theagarayan et al. [13] showed that in the presence of positive Z(4,0), AR decreases; if it is negative, AR increases. If during accommodation, Z(4,0) remains positive instead of negative, and AR will decrease, leading to an AI.

It is known that changes in ocular aberrations during accommodation are mainly due to changes in lens shape [35,36]. Changes in lens shape during accommodation may be different in some individuals, which may make them more susceptible to the development of AI. Aberrations imply an underlying morphology of the visual system that can cause AI. This different morphology is supported by the fact that the aberrations of the control subjects for a given effective accommodation are completely different from the aberrations obtained for the AI group with a similar effective accommodation, suggesting a different morphology of the lens.

Regarding Z(6,0), most previous studies stated that it tends to become more positive with accommodation [9,10,12]. Similar to what was observed with Z(4,0), the results of the control group were in agreement with the literature, i.e., Z(6,0) became more positive with accommodation. On the other hand, the subjects with AI did not show this tendency. This observation in Z(6,0) is probably due to the same reasons verified with Z(4,0).

As Z(4,0) and Z(6,0) have different behaviours during accommodation in subjects with AI, and they can be used as an indicator or predictor of this type of dysfunction. Furthermore, they can be used for the early detection and prevention of AI.

The changes of off-axis Z(4,0) and Z(6,0) caused by accommodation were not statistically significant between the control and AI groups. Contrarily to what was observed in on-axis, the changes of Z(4,0) with 2.50 D of accommodation follow the same tendency in the control and AI groups, i.e., Z(4,0) became more negative in the most of off-axis areas in both groups. Off-axis Z(6,0) did not show any tendency; in some retinal areas it became more negative and others became more positive. As the central vision plays the most important role in our daily tasks [37], it is expected to have more influence in the control of AR than off-axis vision. However, in this study, off-axis aberrometry was only performed for unaccommodated state and with 2.50 D of accommodation, and only for 11.5° and 23°, which may limit the conclusions.

The optical quality of the eye during accommodation is important for the maintenance of adequate near vision and thus avoiding symptoms such as blurred vision during near vision tasks. The evaluation of the optical quality during accommodation, particularly spherical aberrations, can be important in the diagnoses of AI. Moreover, the findings of this study can help to understand the origin of AI and the reason why some subjects develop this kind of dysfunction. However, it is still a preliminary study, and the results are limited by the reasons described below. It is important to increase the sample size and extend this study to other accommodative dysfunctions.

## 5. Conclusions

The changes of on-axis spherical aberrations with accommodation seem to be different in subject with and without AI. In the control group, Z(4,0) became more negative with accommodation and Z(6,0) became more positive, whereas subjects with AI did not show the same tendency. There do not seem to be any differences in off-axis spherical aberrations with accommodation between the subjects with and without AI. Central Z(4,0) and Z(6,0) might be used as an AI indicator and may explain its origin. However, the sample size must increase.

**Author Contributions:** Conceptualization, J.G. and S.F.; methodology, J.G., K.S. and S.F.; software, S.F.; validation, J.G., S.F.; formal analysis, J.G. and S.F.; investigation, J.G., K.S. and S.F; resources, J.G. and S.F.; data curation, J.G. and K.S.; writing—original draft preparation, J.G.; writing—review and editing, J.G. and S.F.; visualization, J.G., K.S. and S.F.; supervision, S.F.; project administration, S.F.; funding acquisition, J.G., K.S. and S.F. All authors have read and agreed to the published version of the manuscript.

**Funding:** This work was supported by the Portuguese Foundation for Science and Technology (FCT) in the framework of the Strategic Funding UID/FIS/04650/2019 and by the project PTDC/FISOTI/ 31486/2017 and POCI-01-0145-FEDER-031486. The author Jessica Gomes is also supported by the PhD grant 2020.08737.BD from FCT.

**Institutional Review Board Statement:** The study was conducted in accordance with Declaration of Helsinki, and was approved by the Ethical Sub commission of Life and Health Science of University of Minho.

**Informed Consent Statement:** All subjects signed an informed consent with the explanation of the procedures.

**Data Availability Statement:** Due to privacy and ethical restrictions, data is not publicly available.

**Conflicts of Interest:** The authors declare no conflict of interest.

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
