# Peer review of "Spherical Aberration and Accommodative Insufficiency: Is There a Link?"

_photonics, doi:10.3390/photonics10040452_

Round 1
Reviewer 1 Report
Review as pdf.

Author Response
We would like to thank the reviewer for the efforts and valuable time to evaluate our manuscript. The points addressed by reviewer allowed us to improve our work.
Please find the point-by-point answer attached.

Reviewer 2 Report
The manuscript is well written. The foundations and experimental development are sufficiently clear and allow easy following of the text-
Given that the set-up is experimental and not a commercial system, I would recommend that the authors add a diagram or a picture of the setup. Additionally, I would add a table with the introduced and effective accommodation data so that direct comparison is possible.
Regarding this, I found an issue in line 128, where the authors comment that the maximum accommodation has been measured only in 17 and 10 subjects when there are only 10 and 11 subjects in each group. I assume it is a typographical error and it refers to 7 and 10 subjects, respectively
Another detail to consider is the wording of the paragraph in lines 204-208, where it seems to suggest that aberrations cause AI. From a physical point of view, aberrations are a way of describing the effect of an optical system on light, but they are not a physical element. Writing it this way seems like an abuse of language, and I believe it would be more correct to say that aberrations imply an underlying morphology of the visual system that causes AI. This different morphology is supported by the fact that the aberrations of the control subjects for a given effective accommodation are completely different from the aberrations obtained for the AI group with a similar effective accommodation, suggesting a different morphology of the lens. This conclusion is actually outlined in the text, and I think that this conclusion should be more stressed.
In any case, I think this is a good work, and I recommend its publication with minor corrections.
Author Response

(The authors gave the same response as above.)

Round 2
Reviewer 1 Report
The manuscript has been corrected by the authors following the sugesstions performed. It can be accepted for publication in the current form.